# Rectal Cancer Tissue Lipidome Differs According to Response to Neoadjuvant Therapy

**DOI:** 10.3390/ijms241411479

**Published:** 2023-07-14

**Authors:** Salvador Sánchez-Vinces, Gustavo Henrique Bueno Duarte, Marcia Cristina Fernandes Messias, Caroline Fernanda Alves Gatinoni, Alex Ap. Rosini Silva, Pedro Henrique Godoy Sanches, Carlos Augusto Real Martinez, Andreia M. Porcari, Patricia de Oliveira Carvalho

**Affiliations:** 1Health Sciences Postgraduate Program, São Francisco University—USF, Bragança Paulista, São Paulo 12916-900, Brazil; salvador.vinces@mail.usf.edu.br (S.S.-V.); gustavo_duarte95@hotmail.com (G.H.B.D.); marcia.cfmessias@gmail.com (M.C.F.M.); carolgatinoni@gmail.com (C.F.A.G.); 2MS4Life Laboratory of Mass Spectrometry, Health Sciences Postgraduate Program, São Francisco University—USF, Bragança Paulista, São Paulo 12916-900, Brazil; alex.rosini@mail.usf.edu.br (A.A.R.S.); pedrohgodoys@gmail.com (P.H.G.S.); andreia.porcari@usf.edu.br (A.M.P.); 3Department of Colorectal Surgery, São Francisco University—USF, Bragança Paulista, São Paulo 12916-900, Brazil; carlos.martinez@usf.edu.br

**Keywords:** rectal cancer, lipidomics, response to neoadjuvant therapy

## Abstract

Rectal cancer (RC) is a gastrointestinal cancer with a poor prognosis. While some studies have shown metabolic reprogramming to be linked to RC development, it is difficult to define biomolecules, like lipids, that help to understand cancer progression and response to therapy. The present study investigated the relative lipid abundance in tumoral tissue associated with neoadjuvant therapy response using untargeted liquid chromatography–mass spectrometry lipidomics. Locally advanced rectal cancer (LARC) patients (n = 13), clinically staged as T3–4 were biopsied before neoadjuvant chemoradiotherapy (nCRT). Tissue samples collected before nCRT (staging) and afterwards (restaging) were analyzed to discover lipidomic differences in RC cancerous tissue from Responders (n = 7) and Non-responders (n = 6) to nCRT. The limma method was used to test differences between groups and to select relevant feature lipids from tissue samples. Simple glycosphingolipids and differences in some residues of glycerophospholipids were more abundant in the Non-responder group before and after nCRT. Oxidized glycerophospholipids were more abundant in samples of Non-responders, especially those collected after nCRT. This work identified potential lipids in tissue samples that take part in, or may explain, nCRT failure. These results could potentially provide a lipid-based explanation for nCRT response and also help in understanding the molecular basis of RC and nCRT effects on the tissue matrix.

## 1. Introduction

Rectal cancer (RC) is one of the top four most deadly cancers worldwide [1]. In the last three decades, there has been a reduction in the incidence of RC and a decrease in its mortality, possibly due to both diagnostic/therapeutic improvements and secondary prevention in high-income countries. However, a constantly increasing incidence in middle- and low-income countries has been observed, underscoring the current importance of this pathology. Clinical characteristics of RC include a change in gastrointestinal habits (i.e., diarrhea, constipation), presence of hematochezia, rectal tenesmus, abdominal pain and systemic symptoms such as iron-deficiency anemia, weight loss, and weakness. Effective coordination among healthcare professionals is required for RC management due to the interdisciplinary nature of its treatment. A multidisciplinary team of oncologists, surgeons, radiotherapists, radiologists, pathologists, and endoscopists is necessary for an optimal oncological outcome [2,3], as shown by a study analyzing unsuccessful multidisciplinary discussions as predictive factors for positive resection margins, as well as the absence of radiotherapy [4]. In medical research, one of the most important aspects is the establishment and development of better therapeutic options for patients. Precise and personalized medical strategies based on the specific metabolite or molecular signatures and cellular context are important to take account of individual variability in cancer metabolism [5]. The characteristics of the mass spectrometry (MS) analytical technique has become the standardized basis of oncological approaches, identifying therapeutically significant components in clinical laboratories, such as endogenous or exogenous molecules and their metabolites, proteins profiling, etc. [6].

When rectal cancer is diagnosed as locally advanced (LARC) (cT3–T4, with perirectal fat or adjacent structure invasion, respectively), or with metastatic perirectal lymph nodes (N+), multimodal strategies with preoperative chemoradiotherapy become crucial for optimizing results. The combination of neoadjuvant chemoradiotherapy (nCRT) and total mesorectal excision (TME) has become the standard option for LARC, highlighting the importance of a multidisciplinary evaluation [4]. Currently, there are no molecular markers in rectal cancer available that can evaluate specific situations or treatments such as when a patient needs preoperative treatment for a localized or locally advanced rectal cancer, indicating that the surgery will not be radical. Similarly, there are no clinically relevant markers that can predict the response to radiotherapy (RT) or CRT [7].

5-Fluorouracil (5-FU) is currently used as the standard chemotherapeutic agent for locally advanced rectal cancer nCRT, while additional chemotherapeutic agents, including capecitabine and oxaliplatin, have shown promise in increasing the complete pathologic response (pCR) compared to the regimen using 5-FU [8] alone. However, the response to nCRT in locally advanced rectal cancer varies among patients. A partial response is present in ~40% of patients and 8–20% of patients reach a pCR at the time of surgery, while some tumors (~20%) exhibit resistance to nCRT (progression or minimal regression to stable disease) [8]. Moreover, given the observation of pCR in a significant proportion of patients undergoing nCRT and to avoid the adverse effects of surgery, alternative approaches such as “watch and wait” or local transanal excision have been suggested [8]. Therefore, there is a critical need to understand the response to nCRT thereby enabling early selection of patients who would or would not benefit from nCRT. Some pathological characteristics have been studied as predictors of the response to nCRT: tumoral differentiation, circumferential tumor, mucosal histology and macroscopic ulceration are associated with poor response to nCRT [9]. Moreover, imaging methods, like positron emission computed tomography, magnetic resonance imaging and endoscopic ultrasound are used for pretreatment staging, evaluation of nCRT response and recovery after nCRT. The tumor regression rates and circumferential resection margin, defined by imaging techniques, can potentially predict the response to nCRT in rectal cancer [9]. However, the usefulness of these clinicopathological and radiological characteristics is currently limited due to the low sensitivity and specificity of rectal cancer [7].

The present work proposed to evaluate the changes in the lipidome of tissue from patients with adenocarcinoma before and after the treatment with neoadjuvant CRT, looking for changes in molecular signatures that can help in understanding the tumor response to treatment.

## 2. Results

### 2.1. Baseline Characteristics of Patients

Table 1 shows the baseline characteristics of participant patients. Values are presented by groups. All patients were staged as T3–4 before nCRT.

### 2.2. Feature Selection

After data preprocessing, statistical analyses were performed. First, an exploratory analysis using PCA plotting shows the fitted cluster formed by the QC samples and the dispersion of the experimental samples (Figure 1a for negative mode and Figure 1b for positive mode). Preprocessed data can be found in Appendix A for abundance in positive ionization mode and in Appendix A for negative ionization mode.

After exploratory analysis, the groups were evaluated using the difference in abundance expressed by their fold-change and the *p*-value obtained by applying the limma model. In the Responders vs. Non-responders comparison before nCRT, 20 features were selected from the negative mode and 47 from the positive mode. In the Responders vs. Non-responders comparison after nCRT, 43 features were selected from the negative mode and 537 from the positive mode.

Annotations were then proposed for those features with enough evidence according to the confidence level mentioned in the methods. Thirty-two features were annotated for the comparison before nCRT and 424 features for the comparison after nCRT.

Additionally, the study used the Lipid Ontology tool (Lion/web) [10] to facilitate the interpretation of these annotated lipid profiles. This method maps the annotated lipids by exact match or similarity defined by the tool itself and compared with databases of biological pathways, functions, or structures. The result is a list of suggestions in which the lipids in question may be participating and then ranked according to the FDR q-value and the sign of the effect.

Table 2 and Table 3 show the list of features selected and annotated for each comparison, mass error and the statistical results.

Appendix A contains the detailed information of the complete list of features obtained from the analysis (such as fragment matches, ion mode, adducts and annotation scores) of the tissue samples and the complete statistical report from limma analysis results.

Table 4 shows the top ten enrichments using the relevant (adj.*p*-value < 0.05) annotated lipids from the Responder vs. Non-responder comparison after nCRT. The Term ID is the ID defined by LION; the Description is the mapped/annotated lipid name; the Annotated column informs the number of lipids from the query present in the enrichments. The *p*-value is the statistical significance and FDR q-value is the corrected *p*-value. The column “Regulated” indicates whether the suggested term is more represented (UP) or less represented (DOWN) in the Responder than in the Non-responder group.

## 3. Discussion

In this study, we focus on tissue lipids endeavoring to detect a discriminatory profile or specific molecules capable of identifying patients who do not respond to nCRT. Biomarkers for nCRT response in LARC could personalize treatment strategies to improve response rates and survival outcomes. Despite the small sample size analyzed, our results indicate 67 relevant annotated lipids that drive the difference between Non-responders and Responders. Figure 2 summarizes the results of the most differential lipids. The results suggest a high confidence level (adjusted *p*-value < 0.05) list of features that discriminate between tissue samples from patients who are Responders to nCRT when compared to Non-responder patients after nCRT (T1) with a suggested identification of 35 lipids. When comparing the same groups before nCRT (T0), a list of 32 selected features was obtained with raw *p*-values < 0.05 and log2FoldChange > 2 and suggested identifications.

The glycerophospholipids are the most abundant phospholipids and the major component of cell membranes. Their glycerol backbone can be esterified by fatty acids with different functional heads, such as choline, ethanolamine or glycerol. These residues can be attached to the glycerol by an O-acyl, or O-alkyl, or O-alk-1′-enyl (plasmalogen). Among these lipids, some glycerophosphoglycerols (PG) are significantly increased in the samples of the Non-responder group, with three ether PG. Plasmanyl phospholipids, also known as plasmalogens, have shown increased levels in colorectal cancer cells when compared with non-tumoral cells [11,12,13], and together with other ether lipids, have been related to the pathogenesis and aggressiveness of cancer [13]. These ether lipids have various biological functions. They serve as a reservoir for second messengers and other biological effects associated with their ability to regulate ion channels that control cell physiology and the reduced expression of multi-drug resistance genes (like *MDR1*, *MRP1* and *ABCG2*) when alkylglyceronephosphate synthase (AGPS) is silenced. Their products (such as lysophosphatidic acid-ether) have been found to be reduced in cancer cell lines [14]. However, it is not clear whether these lipids have a structural or a signaling role in cancer progression. There were increased levels, in Non-responder samples, of ether lipids in other glycerophospholipids, such as phosphocholine (PC O-42:9), phosphoethanolamine (PE O-38:3), phosphoinositols (PI O-36:2) and phosphoserines (PS O-36:1, PS O-38:1, PS O-36:2). The diacylglycerophospholids are some of the lipids more identified in our results and they were more abundant in the Non–responder group. Comparing response after nCRT, it was found that four compounds were more abundant in Responders: two of these were lysophosphoethanolamines (monoacylglycerophospholipids LPE 20:3 and LPE 18:3) and the others were lactosamine and one diacylglycerol (DG 40:6). While increased levels of lysophospholipids as a group have been related to cancer progression [15,16], a previous colorectal cancer study found an increased level of LPC [17], probably due to differences in chain length or the number of unsaturations. More studies are needed to determine the role of polyunsaturated LPE, or other lysophospholipids, in cancer progression and response to therapy. When analyzing results before nCRT, an increased level of some diacylglycerophospholipids (PE 36:0, PC 31:1, PG 34:1 and PG 36:1) was detected in the tissue samples of Responders to nCRT, and a decreased level of monoacylglycerophospholipids (a cardiolipin CL 36:4 and the LPLs LPI 18:0, LPC 18:2, LPE 22:0, LPC 17:2) in the same samples.

The glycosphingolipid GalCer, a HexCer from the Simple Glc series lipid subclass, is more abundant in Non-responders after nCRT. Correspondingly, GlcCer is more abundant before nCRT for Non-responders. This more abundant lipid has been related to the protection of the apoptosis attenuation of Cer-mediatic signals [18,19]. These groups transactivate multidrug resistance 1/P-glycoprotein (*MDR1*) and multidrug resistance-associated protein 1 (*MRP1*) expression, stimulating drug efflux [20]. In a study of lung cancer cells, the enzyme glucosylceramide synthase was found to increase after chemotherapy suggesting that other glycolipids could be involved in drug resistance [21]. Lipids such as lysophosphatidic acid and ceramides influenced the length of telomeres and replicative immortality, therefore being involved in enabling replicative immortality [22]. Among the annotated lipids, we identified some ceramides without a unique subclass (Cer 42:0;O3, Cer 42:0;O3 or Cer 42:1;O3) as being more abundant in Non-responder patients before nCRT, but not after nCRT. The promotion of telomerase activity controls cancer replication since telomere shortening induces senescence [23]. The variability of composition, length and unsaturation of ceramides in cancer has been reported in other studies [22].

Other lipid classes highly represented in samples of the Non-responder group are the oxidized species of PC, PS and PE. Most of the annotated oxidized lipids have the oxidized PUFA residual, probably as a product of alkenyl glycerophospholipids oxidation [24]. There was only an observable difference before nCRT for one oxidized phosphatidylcholine. The presence of oxidized glycerophospholipids has been related to cellular homeostasis and disease progression. Recent studies have found the effect of an increased abundance of these lipids in the tumoral microenvironment regulating autophagy and inducing metastasis [25]. Interestingly, there are reports of the presence of oxidized PS on the surface of apoptotic cells [26], showing the necessity of acquiring more knowledge about the sources and functions of different lipid oxidation products in cancer. It is important to note that the differential oxidized lipid profile as a product of the response to nCRT differs from those reported by recent studies of colorectal cancer where tumor tissue lipidome was compared to adjacent non-tumor tissue lipidome but maintains a similarity to other classes such as sphingolipids and some of the glycerophospholipid subclasses [27,28].

Although the discussions presented here may be useful to better understand the effect of the neoadjuvant treatment on rectal cancer tissue, according to patient’s clinical response, there are some limitations that should be considered. The limited amount of clinical information and the low number of patients reinforce the need for further validation of this experiment in an independent cohort. The dependence on confirmation of the suggested lipid annotation by targeted analysis, and their target quantitation would also add value to these findings. Despite these limitations, these results may help in planning and developing more comprehensive studies.

## 4. Materials and Methods

### 4.1. Study Groups and Sample Collection

Samples were obtained from 13 patients with locally advanced rectal cancer, of both sexes, aged 48 to 83, participating in an observational, analytical study with a prospective collection. The neoadjuvant treatment was performed according to the hospital protocol [29]: 5040 cGy (25 fractions for 6 weeks) and leucovorin (20/mg/m^2^/day) with doses of 5-fluorouracil administered intravenously at 425 mg/m^2^/day for three consecutive days on the first and last three days of radiation therapy. After 6 to 8 weeks, a surgical sample was collected during restaging. There was no interference with the hospital’s clinical management.

Tumoral rectal tissue was collected during the diagnostic biopsy procedure before nCRT (T0) and the surgical procedure after the completion of nCRT (T1).

Sample collection was carried out at the Hospital Universitário São Francisco na Providência de Deus (Bragança Paulista, SP, Brazil). Patients were included after signing the Informed Consent Form approved by the Ethics and Research Committee (CEP) of the Universidade São Francisco (CAAE: 14958819.8.0000.5514).

To evaluate the effect of the nCRT, two unpaired comparisons were carried out in tissue samples: in the first one, lipidome from tumor tissues collected at staging (labeled as T0) of patients with some degree of TNM stage reduction (tumor downstaging) after having received nCRT, labeled as Responders, was compared with that of the group of patients with no tumor downstaging, labeled as Non-responders; in the second comparison, lipidome from tumor tissues collected at restaging (labeled as T1) of patients with some degree of TNM stage reduction (tumor downstaging) after having received nCRT, labeled as Responders, was compared with that of the group of patients with no tumor downstaging, labeled as Non-responders.

### 4.2. Matrix Extraction

#### Tissue Extraction

Liquid extraction was performed for fresh–frozen tissue collected in dry tubes. After thawing to room temperature, tissue sample metabolites were extracted using 500 µL of a cold solution of methanol–H_2_O (4:1, *v*/*v*) and three liquid nitrogen freeze-thawing cycles. Samples were submitted to ultrasound (8 min), centrifuged (9000 rpm, 10 min, 4 °C), and then dried under N_2_ (g) flow. Samples were then resuspended in acetonitrile-water (1:1, *v*/*v*). Due to the tissue’s weight variation, samples with ≤5 mg were resuspended in 180 μL of the resuspension solution. Samples weighing >5 mg had their resuspension volume corrected by the tissue mass to achieve a minimum concentration of 30 mg mL^−1^. Samples used for quality control (QC) were acquired by aliquoting the resuspended samples to form a pool and then divided into 11 replicates and injected during batch acquisition after every 10 sample arrays for system suitability and instrumental variability evaluation.

### 4.3. LC–MS/MS Analysis

The untargeted analysis was performed using an ACQUITY UPLC H-Class (Waters, Manchester, UK) coupled to a XEVO-G2XS QToF Mass Spectrometer (Waters, Manchester, UK). Mobile phase A was composed of a solution of Acetonitrile:H_2_O (60:40, *v*/*v*) with 1% ammonium formate, while mobile phase B was composed of Isopropanol:Acetonitrile (90:10, *v*/*v*) with 1% of ammonium formate. Additionally, an Acquity UPLC CSH C18 column (2.1 × 100 mm, 1.7 μm, Waters, Manchester, UK) focusing on non-polar compounds was used.

The flow rate was 0.4 mL min^−1^. The column was initially eluted with 40% B, increasing to 43% B over 2 min and subsequently to 50% within 0.1 min. Over the next 9.9 min, the gradient was further ramped to 54% B, and then to 70% of B in 0.1 min. Over the next 5.9 min, the gradient was further ramped to 99% B, and then to 40% of B in 0.1 min. The %B was kept for 1.9 min to stabilize the chromatographic column. The total run time was 20 min.

Positive (+) and negative (−) ion modes were recorded (separately) and the instrument was operated in MS^E^ mode in the *m/z* range of 50–1700 *m/z*, with an acquisition time of 0.5 s/scan. The injection volume was 3 μL (+) and 5 μL (−). The source temperature was set to 140 °C (+ and −) and the desolvation temperature to 550 °C (+ and −). The desolvation gas flow was 900 L/h^−1^ (+ and −) in a capillary tension of 3.0 kV (+) and 2 kV (−), with a cone voltage of 40 kV (+ and −). The MS^E^ analysis was operated at 6 V for low collision energy and a ramp of 20–50 V for high collision energy. Sample injection order was random, and QC samples were added intra-batch after each array of ten samples and were also included inter-batch (before and after each batch of analysis).

Leucine enkephalin (molecular weight of 555.62; 200 pg μL^−1^ in an Acetonitrile:H_2_O (1:1 *v*/*v*) solution) was used as a lock–mass for accurate mass measurement.

### 4.4. Data Analysis

Spectrometric signal processing and feature annotation were performed using the Progenesis™ QI software v2.4.69.11 (Nonlinear Dynamics—Newcastle, UK). These processes were carried out at an in-house station configured with an Intel^®^ Core™ i9-9900K CPU processor (Santa Clara, CA, USA) running at 3.60 GHz under a Windows 10 Enterprise Operational System, equipped with 64 GB of RAM and NVIDIA Quadro^®^ (Santa Clara, CA, USA).

The relative abundance data of the detected ions were processed prior to statistical analysis. A signal correction was performed using a method based on the consistency of QC samples (Random Forest QC Signal Correction—RF-QCSC) implemented in the statTarget package v.3.17 [30]. Then, features with low abundance variation were removed using the interquartile range (IQR). Finally, abundances were normalized using square root transformation and Pareto scaling. Filtering, normalization and statistical analysis were performed using the MetaboAnalystR package v.3.3 [31] and limma package v.3.17 [32] implemented in the R programming language v.4.3 using Rstudio IDE v.2023.06.0+421 [33].

Lipid annotation was assigned according to Annotation Confidence Level [34] level 2 of metabolite identification (exact mass, isotopic pattern, retention time, and MS/MS spectrum matched to an in-house spectral database or literature spectra). The study adopted the lipid nomenclature and classification suggested by the LipidMaps consortium [35].

### 4.5. Statistical Analysis

Differences between sample groups were evaluated using the limma model [32], a linear model initially developed for microarray data that enables flexible modeling and increased statistical inference using techniques such as empirical Bayes. A list of relevant features was obtained using a *p*-value of <0.05 and a fold-change between abundances (FC) of >2 as selection criteria. Additionally, an adjusted *p*-value of <0.05 was considered to increase the confidence level of the discussion for some of the results.

## 5. Conclusions

These findings indicated that lipidome derived from tissue could potentially be helpful in identifying patients with LARC who would not respond to nCRT and could help to understand physiological differences between responders and Non-responders to nCRT. Our results suggest that some lipid classes are involved in the resistance to nCRT (like the simple glycosphingolipids) and are in agreement with previous studies about CRT response in cancer therapy. Finally, it was evident that subclasses of lipids maintain an abundance rate between groups at different moments of nCRT, denoted in the oxidation and the type of residues of glycerophospholipids. More studies are needed to validate the identification and function of the lipids mentioned in our results.

## Figures and Tables

**Figure 1 ijms-24-11479-f001:**
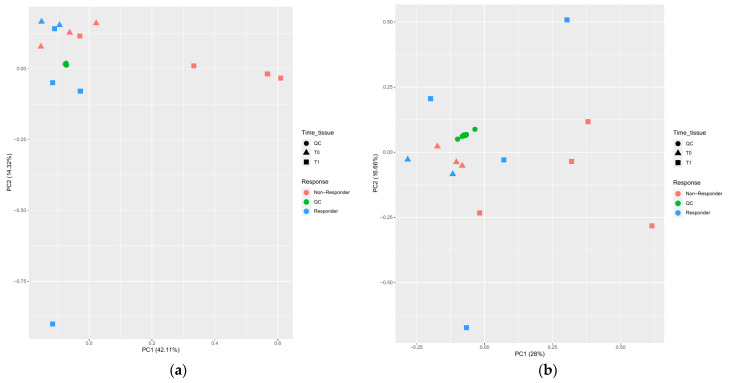
PCA plots for samples after preprocessing data. Features obtained from (**a**) negative ionization mode and (**b**) positive ionization mode for tissue samples. The shape of the data points represents the moment of the analysis (T0 for pre-nCRT and T1 for post-nCRT). Colors represent the response of patients. The quality control (QC) samples are represented by a specific color and shape.

**Figure 2 ijms-24-11479-f002:**
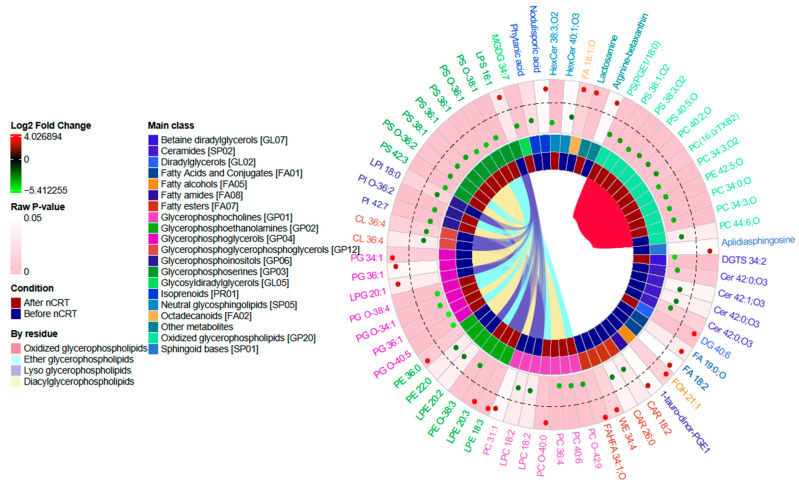
A circular plot of the statistically relevant lipids (cells) when comparing Responders vs. Non-responders. Each ring of the circle represents information on selected lipids: the outer area for fold change results in a green-red scale and a background color representing the *p*-value; the middle area represents the main class for each suggested lipid and the moment of analysis of the sample. The inner area links cells that form part of a specific subclassification by residue. Different cells with the same short annotation represent different features with the same suggested identification.

**Table 1 ijms-24-11479-t001:** Baseline characteristics of patients grouped by the response to nCRT.

	Non-Responders	Responders	*p*-Value ^2^
Subjects	6	7	-
Sex			
F	1	3	0.31
M	5	4	
Age ^1^ (years)	65.3	68.2	0.90
BMI ^1^	26.5	28.8	0.49
cTNM staging			
T3–4	6	7	-
N0	1	1	-
N1–2	5	6	-
M0	6	7	-
M1	0	0	-
ypTNM			
T1–2	0	2	-
T3–4	6	5	-
N0	0	4	-
N1–2	6	3	-
M0	5	7	-
M1	1	0	-

^1^ Mean value by group. ^2^
*p*-value of *t*-test for quantitative values, and Chi-squared test for categorical data, when comparing by response group. BMI: body mass index. F: female. M: male.

**Table 2 ijms-24-11479-t002:** Relevant annotated ions selected by statistical analysis (positive and negative ion mode) of the comparison of samples from Responder vs. Non-responder patients before nCRT.

Feature	Short Annotation	Mass Error (ppm)	logFC	*p*-Value
6.86_536.4819 n	FAHFA 34:1;O	2.7717	4.0269	0.0044
14.65_668.6521 *m/z*	Cer 42:0;O3	−4.5356	−3.9227	0.0055
11.53_792.5733 *m/z*	PE 36:0	−3.6046	3.5604	0.0070
8.73_747.5194 *m/z*	PG 34:1	1.6196	3.4809	0.0084
3.58_924.5146 n	CL 36:4	1.8969	−3.3525	0.0111
15.23_667.6479 n	Cer 42:0;O3	0.0813	−3.3483	0.0112
2.06_297.2431 *m/z*	FA 18:1;O	−1.4135	3.3013	0.0124
6.84_542.4942 *m/z*	WE 34:4	2.1087	3.4591	0.0144
1.15_599.3194 *m/z*	LPI 18:0	−1.2720	−3.2304	0.0144
2.10_352.3573 *m/z*	FOH 21:1	−0.3582	3.4540	0.0145
1.21_330.3361 *m/z*	Phytanic acid	−1.6217	−3.2296	0.0223
4.09_534.2232 *m/z*	LPS 16:1	0.6332	3.0313	0.0320
1.69_537.3200 *m/z*	LPG 20:1	0.3560	2.8287	0.0321
1.82_427.3894 *m/z*	CAR 18:2	−0.0624	−3.0241	0.0324
1.08_434.2429 n	LPC 18:2	−1.0527	−2.8146	0.0330
1.06_564.3305 *m/z*	LPC 18:2	−0.3987	−2.8109	0.0332
3.35_924.5148 n	CL 36:4	2.0625	−3.0082	0.0333
1.34_356.3151 *m/z*	FA 19:0;O	−2.4826	3.0061	0.0334
14.42_923.6832 *m/z*	PC 44:6;O	−1.7134	−2.9993	0.0338
1.54_532.3411 *m/z*	PE 22:0	0.4703	−2.7921	0.0344
15.03_728.6416 *m/z*	Cer 42:0;O3	0.8472	−2.7354	0.0382
1.13_504.3094 *m/z*	LPE 20:2	−0.2464	−2.7346	0.0383
4.86_578.4560 *m/z*	CAR 26:0	2.7616	−2.8895	0.0409
14.68_665.6321 n	Cer 42:1;O3	−0.1767	−2.6842	0.0420
4.93_818.5918 *m/z*	PG 36:1	1.5637	2.8517	0.0436
11.70_716.5240 *m/z*	PC 31:1	0.6475	2.6554	0.0442
9.01_560.5017 *m/z*	Arginine-betaxanthin	0.7099	2.8436	0.0442
2.94_702.3752 *m/z*	Nodulisporic acid	−1.9160	2.8164	0.0463
3.30_468.1815 *m/z*	1-tauro-dinor-PGE1	−3.0617	2.6267	0.0465
1.41_411.3572 *m/z*	Aplidiasphingosine	−2.5984	2.8115	0.0467
14.33_844.6528 *m/z*	HexCer 40:1;O3	1.1088	−2.6115	0.0478
2.59_322.2736 *m/z*	FA 18:2	−1.5321	2.7891	0.0485

ppm: parts per million; logFC: logarithm 2 of fold change.

**Table 3 ijms-24-11479-t003:** Relevant (adj.*p*-value < 0.05) annotated ions selected by statistical analysis (positive and negative ion mode) of the comparison of samples from Responder vs. Non-responder patients after nCRT.

Feature	Short Annotation	Mass Error (ppm)	logFC	*p*-Value	adj.*p*-Value
4.93_818.5918 *m/z*	PG 36:1	1.5637	−5.4123	3.76 × 10^−7^	0.0009
4.62_793.5782 *m/z*	PG O-40:5	4.9132	−5.1380	1.41 × 10^−6^	0.0013
5.77_802.5954 *m/z*	PG O-38:4	−0.3161	−4.8490	5.30 × 10^−6^	0.0032
4.71_774.5655 *m/z*	DGTS 34:2	1.3446	−4.6809	1.11 × 10^−5^	0.0040
5.57_776.5799 *m/z*	PG O-34:1	−0.0956	−4.5629	1.84 × 10^−5^	0.0050
4.48_758.5688 *m/z*	PS O-36:1	−0.8144	−4.3578	4.29 × 10^−5^	0.0075
3.25_787.5374 n	PC 34:3;O2	1.3170	−4.3056	5.29 × 10^−5^	0.0081
4.16_772.5496 *m/z*	PS 36:1	1.1028	−4.2985	5.44 × 10^−5^	0.0081
3.97_782.5685 *m/z*	PC 36:4	−1.1802	−4.1594	9.41 × 10^−5^	0.0110
5.09_826.5937 *m/z*	PS O-38:1	0.5798	−4.1530	9.65 × 10^−5^	0.0110
3.91_771.5420 n	PC 34:0;O	0.7265	−4.1264	1.07 × 10^−4^	0.0112
2.08_908.5738 *m/z*	PS 42:3	−4.5221	−4.0815	1.27 × 10^−4^	0.0121
3.86_790.5596 *m/z*	HexCer 38:3;O2	0.2829	−4.0124	1.65 × 10^−4^	0.0132
2.26_845.5431 n	PS 38:3;O2	1.4796	−3.9826	1.84 × 10^−4^	0.0132
4.16_794.5319 *m/z*	PC 34:3;O	1.6877	−3.9650	1.97 × 10^−4^	0.0132
3.65_866.6132 *m/z*	PI O-36:2	1.7662	−3.9411	2.15 × 10^−4^	0.0132
3.51_798.5834 *m/z*	PC 40:6	4.5004	−3.9329	2.22 × 10^−4^	0.0132
3.56_802.5708 *m/z*	PE 42:5;O	−4.4821	−3.8976	2.53 × 10^−4^	0.0143
3.37_842.6042 *m/z*	PC O-42:9	−1.9223	−3.7572	4.19 × 10^−4^	0.0200
1.20_936.5734 n	PI 42:7	0.7129	−3.7415	4.43 × 10^−4^	0.0207
4.06_796.5469 *m/z*	PS O-36:2	0.7371	−3.7077	4.99 × 10^−4^	0.0217
4.09_789.5528 n	PS 36:1	0.9802	−3.6620	5.85 × 10^−4^	0.0241
4.09_782.5677 *m/z*	PS 38:1	−2.1462	−3.6221	6.71 × 10^−4^	0.0251
1.13_786.5164 *m/z*	MGDG 34:7	1.8204	−3.5987	7.27 × 10^−4^	0.0263
1.97_844.5337 *m/z*	PS(PGE1/18:0)	0.2683	−3.5337	9.06 × 10^−4^	0.0301
2.08_849.5698 n	PS 38:1;O2	−3.9149	−3.5339	9.06 × 10^−4^	0.0301
2.26_853.5469 n	PS 40:5;O	0.0096	−3.4565	1.17 × 10^−3^	0.0354
10.17_651.5341 *m/z*	DG 40:6	−0.8063	3.4514	1.19 × 10^−3^	0.0356
4.02_778.5745 *m/z*	PE O-38:3	3.1779	−3.4113	1.36 × 10^−3^	0.0386
2.59_897.6676 *m/z*	PC 40:2;O	−1.7946	−3.3641	1.58 × 10^−3^	0.0411
1.97_542.2622 *m/z*	LPE 20:3	−4.3578	3.3272	1.78 × 10^−3^	0.0443
1.38_498.2568 *m/z*	LPE 18:3	−4.8319	3.2864	2.03 × 10^−3^	0.0478
7.61_782.7139 *m/z*	PC dO-40:0	−1.2526	3.2809	2.06 × 10^−3^	0.0478
2.10_359.1645 *m/z*	Lactosamine	−4.5984	3.2805	2.07 × 10^−3^	0.0478
3.09_847.5560 n	PC(16:0/TXB2)	−1.7813	−3.2521	2.26 × 10^−3^	0.0494

ppm: parts per million; FC: fold change.

**Table 4 ijms-24-11479-t004:** LION analysis results. Lipid ontology enrichment analysis of the features with the suggested annotation.

Term ID	Description	Number of Lipids Mapped by LION	*p*-Value	FDR *q*-Value	Regulated
LION:0000014	glycerophosphoglycerols [GP04]	4	0.00024	0.00907	DOWN
LION:0012009	lipid-mediated signaling	3	0.00037	0.00907	UP
LION:0000011	glycerophosphoethanolamines [GP02]	3	0.00733	0.11972	UP
LION:0000093	headgroup with negative charge	15	0.06723	0.63618	DOWN
LION:0002966	fatty acid with less than 2 double bonds	26	0.07736	0.63618	DOWN
LION:0012080	endoplasmic reticulum (ER)	9	0.0779	0.63618	UP
LION:0000095	headgroup with positive charge/zwitter-ion	7	0.0943	0.6601	UP
LION:0012081	mitochondrion	6	0.12343	0.66542	DOWN
LION:0002949	fatty acid with 19–21 carbons	14	0.13515	0.66542	DOWN
LION:0000467	contains ether-bond	7	0.1358	0.66542	DOWN

## Data Availability

The data presented in this study are available in Appendix A.

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
