# Peer review of "Rectal Cancer Tissue Lipidome Differs According to Response to Neoadjuvant Therapy"

_ijms, 2023, doi:10.3390/ijms241411479_

Round 1
Reviewer 1 Report
Work of Sánchez-Vinces et al. deals with an untargeted lipidomics study aimed at obtaining new information related to rectal cancer (RC) disease and further response to neoadjuvant therapy. Here, the authors present results stem from relatively low number of subjects in each group (7 responders and 6 non-responders) with substantially wide range of ages within each study groups (aged 48 to 83 years). The authors found changes in phospholipid levels and interestingly also oxidized glycerophospholipids were abundant in samples collected after neoadjuvant chemoradiotherapy.
The results presented could provide some further information related to RC. However, I have some questions that I would like the authors to consider and answer before recommending this manuscript for publication.
Here are my more specific comments:
1. The manuscript seems a bit unfinished. For example, there are some reference information missing (page 4, line 118 and on page 11). In addition, reference number 24 is presented (page 11, line 300), but the manuscript contains only 15 references. The manuscript should be more concise, more clearly organized, and the language should be checked by a native speaker.
2. It would be interesting to see some further data (e.g., clinical data) related to these samples and study subjects, if available.
3. Introduction: I would like to see the aims of the study at the end of the introduction chapter.
4. Discussion: In addition, I would like to see the authors speculate limitations of their study (e.g. low number of subjects, extraction solution not suitable extracting non-polar lipids, etc.) at the discussion chapter.
5. Abstract: Could the authors consider the alternative term for upregulation (page 1, line 26). This term is usually used when talking about a gene expression and production of gene products. In addition, the authors should be careful not to over interpret results. Results stem from low number of subjects.
6. Materials and Methods (page 8, lines 172-182): The authors used extraction method for metabolites. The solution of methanol and water (4:1) (nor acetonitrile and water) is not suitable to extract most non-polar lipids (with high recoveries) present in the sample. The choice of extraction solution must be justified.
7. Materials and Methods (page 9, line 189): Please, give more detailed information about the column.
8. Materials and Methods (page 9, chapter 3.4): Please, include a brief description of statistical analysis used (now limited to referencing to Ritchie et al 2015).
9. Results and discussion (table and figure legends): Legends should be self-explaining. Table legend also missing for table 4. I found figure 2 very difficult to read.
10. Results (Table 2): The authors give exact formula for the lipids (e.g. PG(16:0/18:1 9Z). Without specific standards or ion mobility instrumentation it is impossible to fully identify certain lipids or lipid isomers. Please justify this (i.e., position of double bonds in the lipid structures).
11. Results (Table 2): LPA group lipids are not visible in RP chromatograms. I believe this lipid (PA 18:2_0) is an in-source fragmentation product from PC 18:2_0. In addition, these two lipids have same retention times, which further confirms the in-source fragmentation. Please correct this identification.
12. Discussion (page 10 line 252). The reviewer is unable to find data related to ether lipids from reference 3 (Dodaro et al). Could the authors check the reference and correct the right reference if needed.
13. Discussion (page 10 line 275). LPA 18:2 is most likely an in-source product from LPC 18:2 (e.g., Z. Zhao, Y. Xu, J. Chromatogr. B 877 (2009) 3739). Please either validate this finding or remove this compound from the results.
14. Discussion (page 10 line 295). The authors should describe how they have identified the oxidized PUFA residual in the sn-1 position. Only QTOF information is not suitable for this. Were there some standards available to verify this identification?
15. Supplement: Does the table contain only LI 2 (level of identification) lipids? It would be interesting for the readers to make the complete list of detected lipids (at least all statistically significant lipids) with their identification/annotation information (LI information).
16. References: The authors could consider adding some new references to the manuscript, e.g., Josef Ecker et al (2021, doi: 10.1053/j.gastro.2021.05.009).
The language should be checked by a native speaker.
Reviewer 2 Report
Manuscript ID: ijms-2448173
Title: Lipidome in rectal cancer associated with disease and response to neoadjuvant therapy.
Sànchez-Vinces S et al.
The authors investigated the lipids in tissue samples from patients with rectal cancer, providing a lipid-based potential explanation for the non-responders to neoadjuvant therapy.
Despite the small sample size analyzed, the study is well conducted. However, there are issues that need to be addressed.
Comments:
· The authors should check the bibliography which is incomplete. Some references are missing in the text and, therefore the meaning of the information reported is lost.
· Was only tumor tissue analyzed in this study? Lipidomic data from “healthy” mucosa surrounding the tumor could corroborate the results obtained.
Round 2
Reviewer 1 Report
No further comments.
Please consider still to polish the annotations of molecules in tables 2 and 3 in the final version.
Reviewer 2 Report
No other comments